# From Debt to Green Growth: A Policy Proposal

**Tilemahos Efthimiadis [1],* and Panagiotis Tsintzos [2]**

1  Unit for Energy Security, Distribution and Markets, Directorate for Energy, Mobility and Climate, Joint Research Centre, European Commission, P.O. Box 2, 1755 ZG Petten, The Netherlands

2  Department of Economics, School of Economics and Business Administration, University of Thessaly, 28hs Octovriou 78, 38333 Volos, Greece

*  Correspondence: tilemahos.efthimiadis@ec.europa.eu; Tel.: +31-224-565-003

**Abstract:** Concurrent and overlapping crises, such as the COVID-19 pandemic and increases in energy and food costs, are particularly affecting developing and poor countries, in addition to the continuous and growing negative impacts from climate change. In this paper, we develop a policy proposal and show through a model how an implicit debt servicing standstill for a country with large external debt can be combined with a policy for promoting power generation from renewable energy sources, with positive effects for economic growth and for ensuring debt sustainability. Through a typical decentralized competitive general equilibrium endogenous growth model, we show the long term effects of partially redirecting debt payments to green energy infrastructure to facilitate economic growth and achieve a sustained debt-to-output level. The results indicate that a higher percentage of external public debt increases public capital accumulation, resulting in a higher long-run growth rate, albeit with a slower transition to a steady state. Equivalent results are found regarding the level of financial aid provided for public productive capital.

**Keywords:** energy; investments; debt

## 1. Introduction

The majority of countries that are most affected by climate change do not have the fiscal space for scaling up climate-related investments, as stressed in [1]. Furthermore, many authors have found that climate vulnerability is associated with higher sovereign costs of debt (for example see [2]).

In addition to the effects of climate change, developing and poor countries are especially affected by several overlapping crises, such as abrupt increases in energy and food costs, the COVID-19 pandemic, and others (see, for example, [3]).

In this context, sovereign debt relief or standstill programs have been implemented (see [4] for an overview of relevant IMF programs), while many more are being proposed, as in [3]. Key questions for the design and effectiveness of such programs regard their financing and whether they should be accompanied by specific constraints (conditionality). For example, ref. [5] find that sovereign default risk is hump-shaped in the intensity of conditionality in the long-run. Furthermore, ref. [6] finds that debt relief that is substantially based on incentive-compatible conditions promotes fiscal reform and investment and argues for a combination of outright grants and "soft" loans. One should also note that sovereign pre-emptive restructurings are frequent and can result in lower output losses ([7]).

The 2022 Geneva Report on the World Economy [1] presents alternative financing instruments to fund actions that can mitigate climate change, including grants, debt relief, green bonds, and others. In the context of sovereign debt-related instruments, the authors highlight that future debt restructurings should be based on debt sustainability analyses "at explicitly account for the fiscal costs of climate-related expenditures and include enforceable climate conditionality" [8].

In this paper, we develop a formal model to present a policy proposal where a (quasi) debt relief can be used to promote productive public capital while reinforcing economic growth and debt sustainability.

To provide a practical example, we use the context of international creditors supporting a sovereign developing or poor country through a setup where debt payments are redirected to investments on green energy infrastructure in the country. These investments facilitate economic growth and achieve a sustained debt-to-output level. This is in effect our policy proposal.

We develop a typical decentralized competitive general equilibrium endogenous-growth model, extend the standard endogenous growth literature framework by distinguishing internal from external public debt, and add a conditionality clause linking public investment on productive infrastructure associated with the cost of servicing external public debt.

The financial scheme proposed is that a fraction of external debt service is returned to the sovereign government in the form of foreign aid under the binding agreement that this inflow is fully used for investments in green productive public capital. This is in effect the conditionality clause. A typical example are investments in electricity transmission networks that are essential for integrating (new) electricity generation especially from renewable energy sources (RES). It should be noted that transmission system operators usually operate as natural monopolies and are owned by the state. In addition, for new large-scale RES deployment, grid connection costs should be covered by the respective grid operator [9].

These investments can lead the economy to a long-run balanced growth path where the debt as percentage of output remains constant, reducing (if not eliminating) the possibility of a default. We can note here that [10] found that 62% of defaults occurred when output was below its trend.

This policy proposal protects the interests of international creditors who can avoid a default or haircut on the bonds they hold, while also allowing them to promote their (global) green and sustainable growth objectives. For the "recipient" country, productive public investments for green infrastructure are secured, and the threat of a default is avoided, which could also provide other benefits through increased credibility and reduced risk (see, for example, [11]).

As external debt is in effect still serviced, and investments continuously realized, there are no significant lock-ins, especially for future political leaders. In other words, one could envision an annual or biennial evaluation of the scheme while allowing termination by either party without loss of the benefits already realized.

The rest of the paper is structured as follows. In Section 2, we present the sectors of the economy and define the equilibrium and the associated balanced growth path and find the optimality conditions that describe the dynamic evolution of the economy.

Section 3 shows the impacts of the two main parameters. In Section 4, we present numerical simulations, and Section 5 provides a discussion on the main findings of the model, comments on its limitations, and proposes future research. To reduce clutter, throughout the text, we avoid presenting intermediate algebraic analysis, most of which can be found in the appendices, and the remaining analyses are available upon request.

## 2. The Model

### 2.1. The Sectors of the Economy

We consider a typical decentralized competitive general equilibrium endogenous-growth model with three sectors: the household, firms (production), and the government sector. In this model, the engine for endogenous growth is the stock of (productive) public capital, in the spirit of [12,13] and more similar to the work of [14]. More precisely, the setup of the economy is a decentralized closed economy. The main differences from the work of the mentioned authors are that, in our model, the government can issue public debt

domestically and internationally, invests in public infrastructure, and output is positively affected by foreign aid.

### 2.1.1. The Production Sector

The production sector consists of a unique competitive firm that maximizes profits. The production function is as follows:

$$Y(t) = AK(r)^{(1-a)}E(t)^a, \tag{1}$$

where $Y(t)$ denotes output, $K(t)$ is private capital, $E(t)$ is productive public capital, and $t$ is the time period. We assume that labor is inelastic and equal to unity, the parameter $a$ is output elasticity ($1 > a > 0$), and $A > 0$ is total factor productivity.

Profit maximization of the firm yields the rear wage rate $\omega(t)$ and the real interest rate $r(t)$, as

$$\omega(t) = aA\left(\frac{E(t)}{K(t)}\right)^a, \tag{2}$$

$$r(t) = (1-a)A\left(\frac{E(t)}{K(t)}\right)^a. \tag{3}$$

### 2.1.2. The Household

The household sector consists of a representative infinitely lived household that optimally chooses its consumption path and saves through private capital and internal public debt. The representative household allocates consumption over time to maximize lifetime utility:

$$max_{\{C(t)\}}U(C(t)) = \int_0^{+\infty} e^{-\rho t}\frac{C(t)^{1-\sigma} - 1}{1 - \sigma}dt, \tag{4}$$

subject to its dynamic budget constraint:

$$\dot{W}(t) = (1 - \tau(t))(r(t)W(t) + \omega(t)) - C(t), \tag{5}$$

where $W(t)$ denotes wealth, $\dot{W}(t)$ the derivative of wealth over time, $C(t)$ is consumption, $\rho > 0$ is the household's time preference (discount rate), and $\sigma > 0$ is the inverse of the intertemporal elasticity of substitution, and $1 > \tau(t) > 0$ is a flat tax rate. We assume a flat tax rate since a different taxation scheme, such as progressive tax rates, may give rise to unfavorable dynamics (see [15,16]). The household accumulates wealth by investing in two assets, private capital $K(t)$ and government bonds $Bi(t)$ (sovereign internal debt). Thus, the change in wealth over time is:

$$\dot{W}(t) = \dot{K}(t) + \dot{Bi}(t). \tag{6}$$

The government satisfies its dynamic budget constraint through borrowing, that is, by issuing internal and external bonds. The household absorbs all new internal bond issues, and the remainder of its savings goes to private capital.

### 2.1.3. The Government

We do not impose a fiscal rule on the budget of the government, but rather, we let it freely borrow through the dynamic budget constraint and let the level of debt be endogenously determined in this simple economy setup. The dynamic budget constraint of the government (i.e., the debt accumulation identity) is

$$\dot{B}(t) = r_b(t)B(t) - T(t) + S(t) - FA(t), \tag{7}$$

where $FA(t)$ is foreign aid, $B(t)$ is the stock of public debt, $\dot{B}(t)$ is the evolution of debt over time, $T(t)$ denotes total taxes, $S(t)$ denotes government public spending, and $r_B(t)B(t)$ is

$$r_b(t)B(t) = r(t)Bi(t) + r(t)Be(t), \tag{8}$$

where $B(t)$ is total debt, that is, the sum of internal debt $Bi(t)$ and external debt $Be(t)$.

We let the ratio of internal bonds over external bonds be exogenous:

$$\varphi = \frac{Bi(t)}{Be(t)}. \tag{9}$$

The above notation allows us to distinguish the effects across countries with different sovereign debt profiles. In other words, we assume that the level of initial sovereign debt is fixed, but its internal-to-external ratio can differ (e.g., across countries).

Total taxes are equal to

$$T(t) = \tau(t)(r(t)W(t) + \omega(t)). \tag{10}$$

For simplicity and to clearly identify the effects of foreign aid (explained below), we assume that total public spending is directed to productive public capital, such as infrastructure that enables the integration of renewable energy sources in the energy mix (e.g., electricity transmission lines):

$$S(t) = \dot{E}(t) \tag{11}$$

International creditors offer the government an agreement to facilitate green and sustainable growth, and at the same time protect their interests by leading the economy to a long-run balanced growth path where the debt as a percentage of output remains constant, thus reducing the possibility of a default.

International creditors have an incentive to facilitate green and sustainable growth in line with international agreements and other climate and political agendas and to reduce the possibility that the country defaults. A default would mean that the external bonds would not be repaid, or, in the least, would be heavily discounted.

In this context, international creditors return a percentage of interest payments paid to them $(1 > p > 0)$ in the form of direct foreign aid (*FA*) to be exclusively used for investments in green public productive capital (e.g., investments to increase RES production and usage). These investments will lead the economy to a long-run balanced growth path, where the debt as a percentage of output remains constant, reducing (if not eliminating) the possibility of a default.

In our setup, foreign creditors return a percentage ($p$) of interest payments paid to them in the form of direct foreign aid:

$$FA = r(t)pBe(t). \tag{12}$$

Alternatively, $p$ can be thought of as the percentage of international creditors that participate in this foreign aid agreement. Historically, it is often the case that only the official sector participates in such schemes (e.g., debt relief or payment standstills), with no participation from the private sector [3]. In addition, as this is a non-binding voluntary mechanism, it can be stopped at any moment by either party (creditors or national government), and having a flexible return of interest payments ($p$) would allow an annual matching of needs and willingness to contribute.

As the above foreign aid (*FA*) is fully directed to payments for public investments in renewable energy infrastructure, government spending for such infrastructure is

$$S(t) = \dot{E}(t) = r(t)pBe(t). \tag{13}$$

For simplicity, we assume that the depreciation rate of public capital in (13) is zero, as its inclusion would only reduce the tractability of our results without producing useful insights (see [17] for an analysis of the long-run properties of the depreciation rate of public capital). Thus, the dynamic budget constraint of the government (8) can be written as

$$\dot{Bi}(t) + \dot{Be}(t) = r(t)(Bi(t) + Be(t)) - \tau(t)(r(t)W(t) + \omega(t)). \tag{14}$$

### 2.2. Equilibrium Conditions

An equilibrium allocation is defined as an allocation where the firm maximizes profits, both private factors of production earn their marginal product, the household maximizes lifetime utility subject to its intertemporal budget constraint, and the budget constraint of the government is met.

The optimality conditions of the firm are given in Equations (2) and (3). For the representative agent problem (i.e., the household), we obtain the present-value Hamiltonian as

$$H(t) = e^{-\rho t} \frac{C(t)^{1-\sigma}}{1-\sigma} + q(t)[(1 - \tau(t))(r(t)W(t) + \omega(t)) - C(t)]. \tag{15}$$

We assume that a balanced growth path exists (proved later), that is, a balanced growth path that is a long-run equilibrium in which all endogenous variables grow at the same rate.

From the above and after some algebra, which is presented in Appendix A, we derive the growth rate as

$$\gamma = \frac{\dot{C}(t)}{C(t)} = \frac{(1-\tau)(1-a)A}{\sigma} \left( \frac{E(t)}{K(t)} \right)^a - \frac{\rho}{\sigma}. \tag{16}$$

and the below system of equations that characterize our economy:

$$\frac{\dot{c}(t)}{c(t)} = c(t) + \frac{(1-a)A(1-\tau)}{\sigma}\varepsilon(t)^a - \frac{\rho}{\sigma} - \frac{1 - \tau + \varphi - (1-\alpha)\tau\varphi be(t)}{\sigma}A\varepsilon(t)^a = 0, \tag{17}$$

$$\frac{\dot{be}(t)}{be(t)} = c(t) + \left[ \frac{A(\tau(1-\varphi) - \alpha(1 + \varphi(1-\tau)) + (1-\alpha)\tau\varphi be(t))}{1 + \varphi} - \frac{A\tau}{(1+\varphi)be(t)} \right]\varepsilon(t)^a = 0, \tag{18}$$

$$\frac{\dot{\varepsilon}(t)}{\varepsilon(t)} = c(t) + \frac{(1-a)Abe(t)(p(1+\varphi) + \tau\varphi\varepsilon(t)) - A(1 - \tau + \varphi)\varepsilon(t)}{1 + \varphi}\varepsilon(t)^{a-1} = 0. \tag{19}$$

The existence of a unique steady state, with a corresponding unique long-run growth rate of balanced growth path, is proved in Appendix B.

## 3. The Impact of External Debt on Economic Growth

From (16), one can see that the driver of economic growth is the endogenous variable $\varepsilon(t)$. An important parameter is the ratio of internal bonds over external bonds ($\varphi$). Under the financial aid scheme provided to the government by international creditors, the more the government borrows from them, the more it invests in green infrastructure, which leads to more economic growth.

From (9), a decrease in $\varphi$ refers to a relative change in the ratio of internal to external public debt, towards a higher proportion of external debt. We assume that external debt is held by international creditors, who are compensated with the same rate as internal creditors are, as in Equation (8).

Since the economy is on a balanced growth path, the level of external debt increases at the same rate as the economy grows. This means that the outflow regarding interest payments to international holders increases over time as external debt increases. From (13) and (14), it is implied that the level of aid is also increased at the same pace, and the corresponding investments in infrastructure for integrating RES, which in turn increases

the relevant stock of public capital acts as a positive externality, increasing social returns and facilitating economic growth.

In other words, to prove the above analytically it is sufficient to show that $(\partial \varepsilon(t)/\partial \varphi) < 0$, which is shown in Appendix C.

In a similar manner, we analyze the impact of the rate of interest payments ($p$) redirected from foreign creditors to productive public capital. One expects that a higher $p$ will increase the public capital stock of infrastructure, at a higher pace, ceteris paribus other variables. This type of public capital is considered a pure public good, that is, non-rival in its use and non-excludable. It also acts as a positive productive externality, as in [6], that tends to increase output as it accumulates over time, thus, sustaining long-run economic growth. Therefore, increasing $p$ should increase the rate of accumulation of public capital and increase economic growth.

To show this analytically, it is sufficient to show that $(\partial \varepsilon(t)/\partial p) < 0$, as shown in Appendix D.

Regarding the speed of transition to the steady state, in the endogenous growth theory literature, [13] showed that public spending, as a flow variable, can play a role as a positive externality leading to an equilibrium that exhibits log-run and sustained economic growth. While this novel approach had a significant impact on the literature of endogenous growth and successfully explained the tendency of economies to grow over time, this model did not exhibit transitional dynamics (for a discussion, see [18], explaining the dynamics from initial economic conditions toward the path of balanced growth path).

This drawback was addressed through the work of [14], where the authors regarded productive public capital (a stock variable) as the positive externality that plays the role of the engine of long-run growth. They also show how transitional dynamics arise in this setup. Our work is like that of [14], expanding their model to also consider external and internal public debt and a foreign-aid–public-investment financing scheme. These inclusions allow us to explore a particular setup, with different dynamics, since it expands the model to more dimensions while retaining the properties of the key model.

Having shown that an increase in $\varphi$ lowers the long-run growth rate, while an increase in $p$ increases it, it is also of interest to see how the speed of transition is affected by changes in these parameters. We find that, in general, whenever the steady-state long-run growth is higher, the speed of transition is slower, as one would expect given that the economy must reach a higher rate. The analytical proof is provided in Appendix E.

Another important aspect of our analysis is to explore if the proposed institutional setup can lead the economy to a sustained debt-to-output ratio (a standstill debt ratio). We indeed find that an increase in the internal-to-external debt ratio ($\varphi$) leads to an increase in the steady-state debt-to-output ratio. Similarly, an increase in the percentage of interest payments returned as foreign aid ($p$) causes a decrease in the steady-state debt-to-output ratio.

## 4. Numerical Exercises

In the previous sections, we set up the core model, derived its properties, and analytically derived the effect of the main parameters ($\varphi$ and $p$).

We now proceed to conduct a numerical simulation to demonstrate the effects of the main parameters ($\varphi$ and $p$) on growth and debt, using the typical parametric values found in the relevant literature. For this purpose, a more general version of the model is used, where we allow $\varphi$ to vary from 0.01 (a country relying almost entirely on external debt) to 2 (a country that relies mostly on internal debt., We fix all other parameter values as follows: $a = 0.3$, $A = 0.45$, $\rho = 0.12$, $\sigma = 2$, $\tau = 0.35$, and $p = 0.5$. These values are commonly used in the literature; see for example [19,20].

Using these values and expressing the main variables as percentages of output, we numerically solve the system of Equations (17)–(19), along with Equations (3) and (9). The findings are summarized in Table 1, where one can observe that the analytical results are supported. For example, we see that as the ratio of internal to external debt ($\varphi$) increases,

the long-run steady state growth rate ($\gamma$) decreases, and the debt-to-output ratio ($B/Y$) increases.

**Table 1.** Sensitivity analysis with respect to $\varphi$.

| $\varphi$ | c(t) | be(t) | $\varepsilon$(t) | $\gamma$ | E/Y | B/Y |
|---|---|---|---|---|---|---|
| 0.01 | 0.288 | 0.596 | 1.801 | 0.062 | 3.355 | 1.121 |
| 0.11 | 0.295 | 0.562 | 1.721 | 0.061 | 3.250 | 1.177 |
| 0.51 | 0.314 | 0.456 | 1.468 | 0.055 | 2.907 | 1.363 |
| 0.91 | 0.324 | 0.382 | 1.288 | 0.050 | 2.652 | 1.504 |
| 1.31 | 0.330 | 0.328 | 1.153 | 0.047 | 2.455 | 1.616 |
| 1.71 | 0.333 | 0.287 | 1.048 | 0.044 | 2.296 | 1.707 |
| 1.91 | 0.334 | 0.270 | 1.003 | 0.043 | 2.228 | 1.747 |

As a second exercise, we let $p$ vary from 0.01 (almost no foreign aid) to 1 (all interest rates are returned as foreign aid) with a step of 0.1, set $\varphi = 0.5$, and fix all other parameter values as before. The findings are summarized in Table 2, where we observe that the analytical results are supported. As the percentage of interest paid to international creditors that is returned in the form of foreign aid to support public investment in infrastructure ($p$) increases, the long-run steady-state growth rate ($\gamma$) increases, and the debt-to-output ratio ($B/Y$) decreases.

**Table 2.** Sensitivity analysis with respect to $p$.

| $p$ | c(t) | be(t) | g(t) | $\gamma$ | E/Y | B/Y |
|---|---|---|---|---|---|---|
| 0.01 | 0.202 | 0.386 | 0.205 | 0.004 | 0.734 | 2.066 |
| 0.11 | 0.245 | 0.421 | 0.507 | 0.024 | 1.380 | 1.721 |
| 0.31 | 0.288 | 0.446 | 1.019 | 0.043 | 2.252 | 1.478 |
| 0.51 | 0.315 | 0.458 | 1.497 | 0.056 | 2.947 | 1.354 |
| 0.71 | 0.336 | 0.467 | 1.958 | 0.065 | 3.557 | 1.272 |
| 0.91 | 0.353 | 0.473 | 2.409 | 0.073 | 4.112 | 1.211 |

## 5. Discussion

Above, we explored how international creditors can support a developing or poor country using a scheme where debt payments are redirected to investments in green energy infrastructure to facilitate economic growth and achieve a sustained debt-to-output level.

For this purpose, we followed the endogenous growth literature and extended the standard models by including internal–external public debt and foreign aid for productive public capital. We assume that a fraction of external debt service can return to the sovereign government in the form of foreign aid under a binding agreement such that this inflow is fully directed for investing in productive public capital (a conditionality clause).

The key theoretical finding is that this economy has one unique equilibrium, or steady state, that corresponds to a unique balanced growth path. This finding is important as it means that the economy can achieve long-run growth while avoiding any poverty traps, and at the same time achieve a sustained public debt as a percentage of output level.

In this setup, we regarded two important exogenous policy parameters, the internal-to-external public debt ratio and the index of external debt servicing costs that are transformed into productive public capital, that is, $\varphi$ and $p$, respectively. Considering $\varphi$, the intuition is that the government debt portfolio with a higher percentage of external public debt is associated with higher interest payments to international creditors. In a different setup, [21] showed that this may potentially reduce the long-run economic growth.

With the model presented in this document, and considering the (quasi-)conditionality proposed, we show that a higher percentage of external public debt (lower $\varphi$) or higher aid (higher $p$) increases public capital accumulation, resulting in a higher long-run growth rate, while the strong growth properties retain a lower level of aggregate debt to output. The trade-off that occurs is that while achieving a stronger growth perspective for the economy

the transition to the steady state is slower, which is intuitive as more time is needed to reach the higher equilibria.

One can only acknowledge that such macroeconomic models are subtractive from reality. This model is a simple and stylized one, and its findings should be treated with caution. The validity of the theoretical findings should be at least partially supported by further empirical research, an exercise that can prove challenging as the proposed scheme has not (yet) been implemented in practice.

In effect, our paper is meant to serve as a policy proposal, as any scheme would need to be tailored to the specificities of a county (debt profile, green investment potentials, etc.), but also to the risk appetite, preferences, and political realities. However, the proposed structure can inform a support scheme that can be credible and effective, providing benefits for both the creditors (protection of the bonds they hold, achievement of global sustainability goals, etc.) and the recipient country (e.g., avoidance of defaults, credibility for other investments, etc.). Another consideration regards the design of the particular financial mechanism and the relevant institutional set-up. For a similar case, one can refer to [3], which describes the basic mechanics for a temporary debt standstill due to COVID-19. Although their description is at a relatively high level, the text demonstrates how such mechanisms can become quite complicated, requiring specialized experts. For a step-by-step guide to a debt restructuring, see [22].

Another shortcoming is that since the economy is on a balanced growth path, this implies that international creditors should continue and increase their exposure and support to the economy to retain a constant internal–external debt ratio. This observation is not intended to disregard such a financing mechanism, but rather to provoke considerations for an exit strategy following the achievement of the main goals.

Finally, for ease of modeling, we opted to refer to foreign aid. Alternatively, one could envision issuing new government bonds with long maturities for the funds received, taking ownership of the transmission system operator(s), an institutional setup where electricity grid development plans are also approved by foreign creditors, etc. In future work, we plan to extend the model and analysis considering the above.

**Author Contributions:** Conceptualization, T.E. and P.T.; methodology, T.E. and P.T.; software, T.E.; formal analysis, P.T.; resources, T.E.; writing—original draft preparation, T.E.; writing—review and editing, P.T.; visualization, P.T.; project administration, T.E. All authors have read and agreed to the published version of the manuscript.

**Funding:** This work was supported by the Joint Research Centre of the European Commission (WP 2022–2023, Project 32440-PCI).

**Institutional Review Board Statement:** Not applicable.

**Informed Consent Statement:** Not applicable.

**Data Availability Statement:** Not applicable.

**Acknowledgments:** The authors would like to thank Marcelo Masera, Francesco Careri, and participants for the 2022 ISNGI and 2022 IMAEF conferences. The usual disclaimer applies.

**Conflicts of Interest:** The authors declare no conflict of interest.

## Appendix A. Deriving the Balanced Growth Path

Considering the present-value Hamiltonian from the main text, Equation (15), we derive the first-order conditions as

$$\frac{\partial H(t)}{\partial C(t)} = 0 \implies e^{-\rho t} C(t)^{1-\sigma} - q(t), \tag{A1}$$

$$\frac{\partial H(t)}{\partial W(t)} = -\dot{q}(t) \implies r(t)(1-\tau)q(t) = -\dot{q}(t) \tag{A2}$$

The corresponding transversality condition is

$$\lim_{t \to \infty} q(t)W(t), \tag{A3}$$

From Equations (A1) and (A2), we derive the condition of optimal path of consumption as

$$\frac{\dot{C}(t)}{C(t)} = \frac{r(t)(1-\tau)}{\sigma} - \frac{\rho}{\sigma}, \tag{A4}$$

Replacing in (A4) the interest rate from Equation (3) gives us the growth rate:

$$\gamma = \frac{\dot{C}(t)}{C(t)} = \frac{(1-\tau)(1-a)A}{\sigma}\left(\frac{E(t)}{K(t)}\right)^a - \frac{\rho}{\sigma}, \tag{A5}$$

which is also the expression of the long-run growth rate of the economy. We use (9) to transform the rule of motion of public debt (14) into an equation of evolution of external public debt, and with the help of Equations (2), (3), and (6) we derive

$$\dot{Be}(t) = \frac{(1-a)A(1+(1-\tau)\varphi)Be(t) - \tau AK(t)}{1+\varphi}\left(\frac{E(t)}{K(t)}\right)^a \tag{A6}$$

The public investment rule Equation (13) can be written as

$$\dot{E}(t) = (1-a)A\left(\frac{E(t)}{K(t)}\right)^a pBe(t) \tag{A7}$$

From the household's dynamic budget constrain, (5), we can derive to law of motion of private physical capital. We eliminate wealth variable using (6) to express the constraint in terms of physical capital, and we also eliminate internal debt using (9). Finally, we replace the real interest rate and the wage rate from the optimality conditions of the competitive firm, thus, (5) can now be rearranged and written as:

$$\dot{K}(t) = \frac{(1-\tau+\varphi)AK(t) - A(1-a)\tau\varphi Be(t)}{1+\varphi}\left(\frac{E(t)}{K(t)}\right)^a - C(t), \tag{A8}$$

The set of equations that fully describes our economy is

$$\frac{\dot{C}(t)}{C(t)} = \frac{(1-\tau)(1-a)A}{\sigma}\left(\frac{E(t)}{K(t)}\right)^a - \frac{\rho}{\sigma}, \tag{A9}$$

$$\frac{\dot{Be}(t)}{Be(t)} = \frac{(1-a)A(1+(1-\tau)\varphi)Be(t) - \tau AK(t)}{1+\varphi}\left(\frac{E(t)}{K(t)}\right)^a, \tag{A10}$$

$$\frac{\dot{E}(t)}{E(t)} = (1-\alpha)A\left(\frac{E(t)}{K(t)}\right)^a pBe(t), \tag{A11}$$

$$\frac{\dot{K}(t)}{K(t)} = \frac{(1-\tau+\varphi)AK(t) - A(1-a)\tau\varphi Be(t)}{1+\varphi}\left(\frac{E(t)}{K(t)}\right)^a - C(t), \tag{A12}$$

For the solution of the model, we follow [17], as their work uses a similar framework investigating the impact of various fiscal regimes on economic growth. Let us now introduce the following auxiliary variables:

$$\varepsilon(t) = \frac{E(t)}{K(t)},$$

$$c(t) = \frac{C(t)}{K(t)},$$

$$be(t) = \frac{Be(t)}{K(t)},$$

We can now use the above notation and write Equations (A9)–(A12) in terms of growth rates of the corresponding variables:

$$\frac{\dot{C}(t)}{C(t)} = \frac{(1-\tau)(1-a)A}{\sigma}\varepsilon(t)^a - \frac{\rho}{\sigma}, \tag{A13}$$

$$\frac{\dot{Be}(t)}{Be(t)} = \frac{(1-a)A(1+(1-\tau)\varphi)be(t) - \tau AK(t)}{(1+\varphi)be(t)}\varepsilon(t)^a, \tag{A14}$$

$$\frac{\dot{E}(t)}{E(t)} = (1-\alpha)Apbe(t)\varepsilon(t)^{a-1}, \tag{A15}$$

$$\frac{\dot{K}(t)}{K(t)} = \frac{(1-\tau+\varphi)A - A(1-a)\tau\varphi be(t)}{1+\varphi}\left(\frac{E(t)}{K(t)}\right)^a\varepsilon(t)^a - c(t), \tag{A16}$$

If a balanced growth path exists, it is defined as a long-run equilibrium in which all endogenous variables grow at the same rate, or, in this economy:

$$\frac{\dot{C}(t)}{C(t)} = \frac{\dot{W}(t)}{W(t)} = \frac{\dot{T}(t)}{T(t)} = \frac{\dot{B}(t)}{B(t)} = \frac{\dot{Be}(t)}{Be(t)} = \frac{\dot{Bi}(t)}{Bi(t)} = \frac{\dot{E}(t)}{E(t)} = \frac{\dot{K}(t)}{K(t)} = \frac{\dot{Y}(t)}{Y(t)} = \gamma, \tag{A17}$$

where $\gamma$ is positive. Now, we can reduce the dimension of the equation set (A13)–(A16) using (A17) to a three-dimensional system that exhibits a steady state:

$$\frac{\dot{c}(t)}{c(t)} = \frac{\dot{C}(t)}{C(t)} - \frac{\dot{K}(t)}{K(t)} = \gamma - \gamma = 0, \tag{A18}$$

$$\frac{\dot{\varepsilon}(t)}{\varepsilon(t)} = \frac{\dot{E}(t)}{E(t)} - \frac{\dot{K}(t)}{K(t)} = \gamma - \gamma = 0, \tag{A19}$$

$$\frac{\dot{be}(t)}{be(t)} = \frac{\dot{Be}(t)}{Be(t)} - \frac{\dot{K}(t)}{K(t)} = \gamma - \gamma = 0, \tag{A20}$$

which yields Equations (17)–(19) in the main text.

**Appendix B. Existence and Uniqueness**

The equation system (17)–(19) describes the dynamics and the steady state of the economy. The variable that is most crucial in the behavior of the economy is $\varepsilon(t)$ because of the non-linear expressions that appear in the equation and because it determines the steady-state growth rate, or, the balanced growth path growth rate $\gamma$, as given by (16). The system (17)–(19) cannot be solved algebraically, so the issue of existence and uniqueness of a steady state arises (and its corresponding balanced growth path, since each steady state implies a different long-run growth path).

For simplicity, we set $\sigma = 1$, which implies a logarithmic utility function, and solving Equations (17) and (18), we derive the expressions of $c(t)$ and $be(t)$ as functions of $\varepsilon(t)$:

$$c(t) = \frac{\rho^2(1+\varphi)}{\rho(1+\varphi) + (1-a)A\tau\varepsilon(t)^a} + \frac{\rho[\tau(1+\varphi) + a(1-2\tau + (1-\tau)\varphi] + (1-a)aA(1-\tau)\tau\varepsilon(t)^a}{\rho(1+\varphi) + (1-a)A\tau\varepsilon(t)^a}A\varepsilon(t)^a, \tag{A21}$$

$$be(t) = \frac{A\tau\varepsilon(t)^a}{\rho(1+\varphi) + (1-a)A\tau\varepsilon(t)^a}, \tag{A22}$$

We replace (A21) and (A22) in (19), and we derive the law of motion of $\varepsilon(t)$:

$$\frac{\dot{\varepsilon}(t)}{\varepsilon(t)} = \frac{(1-a)A\rho(1+\varphi-\tau(2+\varphi))\varepsilon(t)^a}{\rho(1+\varphi)+(1-a)A\tau\varepsilon(t)^a}$$
$$-\frac{(1-a)A^2\tau\varepsilon(t)^{2a-1}(p-(1-a-(1-a)\tau)\varepsilon(t))}{\rho(1+\varphi)+(1-a)A\tau\varepsilon(t)^a} - \frac{\rho^2(1+\varphi)}{\rho(1+\varphi)+(1-a)A\tau\varepsilon(t)^a} = 0, \quad \text{(A23)}$$

The function on the right hand of (A23) is continuous and differentiable in $\varepsilon(t) \in (0, +\infty)$, as the common denominator $\rho(1+\varphi) + (1-a)A\tau\varepsilon(t)^a$ is positive in $\varepsilon(t) \in (0, +\infty)$.

Taking the derivative of (A23) with respect to $\varepsilon(t)$, we derive the following expression:

$$\frac{\partial F(\varepsilon(t))}{\partial(\varepsilon(t))} = (1-a)A\tau\varepsilon(t)^{a-2}\left[p + a(1-\tau)\varepsilon(t)\frac{p\rho(1+\varphi)(\rho+\rho\varphi)+A\tau\varepsilon(t)^a}{\left(\rho(1+\varphi)+(1-a)A\tau\varepsilon(t)^a\right)^2}\right], \quad \text{(A24)}$$

The first term is positive, while the term in brackets can be rearranged as follows:

$$\frac{\alpha\rho^2(1-\tau)(1+\varphi)^2\varepsilon(t) + (1-a)^2A^2\tau^2\varepsilon(t)^{2a}(p + A\rho\tau(1+\varphi)\varepsilon(t)^a((1-2\alpha)p + 2(1-a)a(1-\tau))\varepsilon(t)}{\left(\rho+\rho\varphi+(1-a)A\tau\varepsilon(t)^a\right)^2}$$

which is also positive in $\varepsilon(t) \in (0, +\infty)$ for $1/2 > a > 0$; see, for example [23], where they use a value of $\alpha$ equal to 0.10 and [24] for a similar analytical parametric restriction and quantitative analysis.

Additionally, for the expression in Equation (A23), one can see that:

$$\lim_{\varepsilon(t)\to 0} F(\varepsilon(t)) = -\infty \quad \lim_{\varepsilon(t)\to\infty} F(\varepsilon(t)) = \infty \text{ for } \frac{1}{2} > a > 0$$

This means that there exists one and unique $\varepsilon(t) \in (0, +\infty)$ that solves (A23) for $1/2 > a > 0$. Therefore, there exists a unique steady state with a unique corresponding long-run growth rate of a balanced growth path.

**Appendix C. Relationships of $\varphi$ and $p$ on Economic Growth**

As discussed in the main text, to prove that a higher $\varphi$ leads to a higher (steady-state) growth rate, it is sufficient to show that $(\partial\varepsilon(t)/\partial\varphi) < 0$.

Recalling from Appendix A that $\varepsilon(t) = \frac{E(t)}{K(t)}$, the growth rate, Equation (16), can be re-written as

$$\gamma = \frac{\dot{C}(t)}{C(t)} = \frac{(1-\tau)(1-a)A}{\sigma}(\varepsilon(t))^a - \frac{\rho}{\sigma}. \quad \text{(A25)}$$

We solve (A23) for $\varepsilon(t)$ and find the sign of $\partial\varepsilon(t)/\partial\varphi$. We observe that the function on the right-hand side of (A23) is an implicit function and cannot be solved analytically, so we resolve the implicit function theorem, and we derive the following expression:

$$\frac{\partial\varepsilon(t)}{\partial\varphi} = -\frac{\partial\left(\frac{\dot{\varepsilon}(t)}{\varepsilon(t)}\right)}{\partial\left(\frac{\dot{\varepsilon}(t)}{\varepsilon(t)}\right)} = \frac{Ap\rho\tau\varepsilon(t)^{1+a}}{\omega_1 + \omega_2 + \omega_3}. \quad \text{(A26)}$$

where

$$\omega_1 = -a\rho^2(1-\tau)(1+\varphi)^2\varepsilon(t) < 0$$
$$\omega_2 = -(1-a)^2A^2\tau^2\varepsilon(t)^{2a}(p + a(1-\tau)\varepsilon(t)) < 0$$
$$\omega_3 = -A\rho\tau(1+\varphi)\varepsilon(t)^a((1-2a)p + 2(1-a)a(1-\tau)\varepsilon(t)) < 0$$

Thus, $(\partial\varepsilon(t)/\partial\varphi) < 0$ is negative for $1/2 > a > 0$.

Similarly, for $p$, it is sufficient to show that $(\partial\varepsilon(t)/\partial p) < 0$. We follow the same steps as above and derive

$$\frac{\partial \varepsilon(t)}{\partial p} = -\frac{\frac{\partial\left(\frac{\dot{\varepsilon}(t)}{\varepsilon(t)}\right)}{\partial p}}{\frac{\partial\left(\frac{\dot{\varepsilon}(t)}{\varepsilon(t)}\right)}{\partial \varepsilon(t)}} = \frac{1}{(1-a)a(1-\tau) + \frac{p(1-a)}{\varepsilon(t)} + \frac{a\rho(1-\tau)(1+\varphi)\varepsilon(t)^{-a}}{A\tau} - \frac{ap\rho(1+\varphi)}{\varepsilon(t)\left(\rho(1+\varphi)(1-\alpha)A\tau\varepsilon(t)^a\right)}}. \tag{A27}$$

We observe that the terms $(1-a)a(1-\tau)$, $\frac{p(1-a)}{\varepsilon(t)}$, and $\frac{ap\rho(1+\varphi)}{\varepsilon(t)\left(\rho(1+\varphi)(1-\alpha)A\tau\varepsilon(t)^a\right)}$
$\frac{a\rho(1-\tau)(1+\varphi)\varepsilon(t)^{-a}}{A\tau}$ are positive, while the term $\frac{ap\rho(1+\varphi)}{\varepsilon(t)\left(\rho(1+\varphi)(1-\alpha)A\tau\varepsilon(t)^a\right)}$ is negative.

Rearranging the terms $\frac{p(1-a)}{\varepsilon(t)} - \frac{ap\rho(1+\varphi)}{\varepsilon(t)\left(\rho(1+\varphi)(1-\alpha)A\tau\varepsilon(t)^a\right)}$, we obtain:

$$\frac{p\left[(1-2a)\rho(1+\varphi) + (1-\alpha)^2 A\tau\varepsilon(t)^2\right]}{\varepsilon(t)\left[\rho(1+\varphi) + (1-a)A\tau\varepsilon(t)^a\right]}$$

which is positive for $1/2 > a > 0$. The above proves that $\left(\frac{\partial \varepsilon(t)}{\partial p}\right) > 0$. Thus, a higher percentage of interest payments on external debt returned as foreign aid to support public investment increases the steady state value of $\varepsilon(t)$, and in turn, it increases the long run growth rate.

**Appendix D. Speed of Transition to the Steady State**

As previously seen, the key endogenous variable that drives the equilibrium results and dynamics of the economy is $\varepsilon(t)$, where the law of motion of $\varepsilon(t)$ is governed by Equation (A23).

If one assumes that initial conditions lay on the dynamic path of (4), then the coefficient of the linear approximation of (A23) with respect to the steady state value of $\varepsilon(t) = \bar{\varepsilon}(t)$ represents the speed of convergence, that is,

$$\frac{\partial\left(\frac{\dot{\varepsilon}(t)}{\varepsilon(t)}\right)}{\partial \bar{\varepsilon}(t)} = (1-a)A\bar{\varepsilon}(t)^{-2+a}\left[p + a(1-\tau)\bar{\varepsilon}(t) - \frac{p\rho(1+\varphi)(\rho + \rho\varphi + A\tau\bar{\varepsilon}(t))}{\rho(1+\varphi)(1+a)A\tau\bar{\varepsilon}(t)}\right] = F(\bar{\varepsilon}(t), p, \varphi). \tag{A28}$$

Therefore, the change in the speed of convergence with respect to $p$ is

$$\frac{\partial(\bar{\varepsilon}(t), p, \varphi)}{\partial p} = -(1-a)A\bar{\varepsilon}(t)^{-2+a}\left[1 - \frac{\rho(1+\varphi)(\rho + \rho\varphi + A\tau\bar{\varepsilon}(t)^a)}{(\rho(1+\varphi) + (1+a)A\tau\bar{\varepsilon}(t))^2}\right]. \tag{A29}$$

The term $-(1-a)A\bar{\varepsilon}(t)^{-2+a}$ is negative, and by rearranging the last term we obtain

$$\left[\frac{A\tau\bar{\varepsilon}(t)\left((1-2a\rho)(1+\varphi) + (1-a)^2 A\tau\bar{\varepsilon}(t)^2\right)}{(\rho(1+\varphi) + (1+a)A\tau\bar{\varepsilon}(t))^2}\right] > 0, \tag{A30}$$

which is positive for $\frac{1}{2} > a > 0$, and thus, (A29) is negative. This means that an increase in $p$ increases the steady-state long-run growth but reduces the speed of transition to the steady state, as one would expect. In other words, a higher $p$ means that the economy requires more time to reach to the steady state, as the (steady-state) debt to output will be lower.

Likewise for $\varphi$, following the same steps as above, one can find that an increase in $\varphi$ decreases the steady-state long-run growth but also increases the speed of transition to the steady state, requiring less time for the economy to reach to the steady state (lower balanced growth path).

**Appendix E. The Impact of $\varphi$ and $p$ on Debt to Output**

As total sovereign debt ($B$) is the sum of internal ($Bi$) and external ($Be$) debt, we can use Equation (9) to express total debt as

$$B(t) = \varphi Be(t) + Be(t). \tag{A31}$$

We divide both sides by the output, as given by Equation (1), to obtain:

$$\frac{B(t)}{Y(t)} = \frac{\varphi Be(t)}{AK(t)^{1-a}E(t)^a} + \frac{Be(t)}{AK(t)^{1-a}E(t)^a}$$

Rearranging the above, utilizing the auxiliary variables $\varepsilon(t) = E(t)/K(t)$ and be $(t) = Be(t)/K(t)$ gives us

$$\frac{B(t)}{Y(t)} = \frac{\varphi be(t)}{A\varepsilon(t)^a} + \frac{be(t)}{A\varepsilon(t)^a}. \tag{A32}$$

Substituting (A22) into the above (A32), we obtain

$$\frac{B(t)}{Y(t)} = \frac{\tau(1+\varphi)}{\rho(1+\varphi) + (1-\alpha)A\tau\varepsilon(t)^a}. \tag{A33}$$

As the existence and uniqueness of a steady state were shown in Appendix B, there is a unique steady-state value of $\varepsilon(t)$; from (A33), one can see that the economy results in a sustained debt-to-output level. Therefore, the economy can reach a steady state that also implies an endogenous level of debt to output.

We now analytically derive how $p$ and $\varphi$ affect the debt-to-output ratio. If we regard the steady-state level of debt to output, then we assume that $\varepsilon(t) = \bar{\varepsilon}(t) = \bar{\varepsilon}(\varphi)$, so we can take the partial derivative (A33) with respect to $\varphi$ to obtain

$$\frac{\partial\left(\frac{B(t)}{Y(t)}\right)}{\partial\varphi} = \frac{(1-a)A\tau^2\varepsilon(\varphi)^{a-1}(\varepsilon(\varphi) - \alpha(1+\varphi)\varepsilon\prime(\varphi))}{\left(\rho(1+\varphi) + (1-\alpha)A\tau\varepsilon(\varphi)^\alpha\right)^2}. \tag{A34}$$

which is positive since $\varepsilon'(\varphi) < 0$ for $1/2 > a > 0$, as can be observed from Equation (A27). This means that an increase in the internal-to-external public debt ratio causes an increase in the steady-state debt-to-output ratio.

Now, if we assume that $\varepsilon(t) = \bar{\varepsilon}(t) = \bar{\varepsilon}(p)$, we can take the partial derivative of (A34) with respect to $p$ to obtain:

$$\frac{\partial\left(\frac{B(t)}{Y(t)}\right)}{\partial p} = -\frac{(1-a)aA\tau^2(1+\varphi)\varepsilon(p)^{a-1}\varepsilon'(p)}{\left(\rho(1+\varphi) + (1-\alpha)A\tau\varepsilon(p)^\alpha\right)^2}. \tag{A35}$$

which is positive as $\varepsilon'(p) > 0$ for $1/2 > a > 0$, from Equation (A27). This means that an increase in the percentage of interest returned to the country as foreign aid causes a decrease in the steady-state debt-to-output ratio.

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
