# Peer review of "From Debt to Green Growth: A Policy Proposal"

_sustainability, doi:10.3390/su15043506_

Round 1
Reviewer 1 Report
In the first introductory chapter, the hypothesis on the basis of which the research is carried out is not set, which is usual when writing research articles.
After the first introductory chapter, a special chapter "Literature review" is missing, which should have presented an overview of previous research conducted by other authors on the same or similar topic.
In the second chapter, "The Model", a theoretical model that has not been tested on real data is presented in a separate chapter.
In the conclusion, the proof of the hypothesis is not explained because it was not stated in the introduction.
The text should have used the Hardware style of literature citation, which is normally used in research papers.
Finally, when citing the literature (References), the number should have been greater than the 14 mentioned.
Author Response
Dear reviewer,
First of all we would like to thank you for your time and efforts.
Your comments have indeed led to an improved paper.
On the particular comments:
- “In the first introductory chapter, the hypothesis on the basis of which the research is carried out is not set, which is usual when writing research articles.” and “In the conclusion, the proof of the hypothesis is not explained because it was not stated in the introduction.”
We state our goal in the introduction “… we develop a formal model to show how a (quasi) debt relief can be used to promote productive public capital can lead to economic growth and debt sustainability.” If it would improve readability we could rephrase the following (from the introduction) in the form of a question/hypothesis.
- “After the first introductory chapter, a special chapter "Literature review" is missing, which should have presented an overview of previous research conducted by other authors on the same or similar topic.”
We have expanded the introduction to include a richer discussion with relevant references.
- “In the second chapter, "The Model", a theoretical model that has not been tested on real data is presented in a separate chapter.”
Indeed, we had already noted in the last section (‘Discussion’) that “The validity of the theoretical findings should be at least partially supported by further empirical research, an exercise which can prove challenging as the proposed scheme has not (yet) been implemented in practice.” and have added “In effect, our paper is meant to serve as a policy proposal, as any scheme would need to tailored to the specificities of a county (debt profile, green investment potentials, etc.), but also to the risk appetite, preferences, and political realities.”
We also note in this response that papers presenting models without accompanying empirical research are common in the relevant literature. Some examples:
- Arellano, Cristina, and Yan Bai. "Renegotiation policies in sovereign defaults." American Economic Review 104.5 (2014): 94-100.
- Chen, Shu-Hua, and Jang-Ting Guo. "Progressive taxation and macroeconomic (In) stability with productive government spending." Journal of Economic Dynamics and Control 37.5 (2013): 951-963.
- Vella, Eugenia, Evangelos V. Dioikitopoulos, and Sarantis Kalyvitis. "Green spending reforms, growth, and welfare with endogenous subjective discounting." Macroeconomic Dynamics 19.6 (2015): 1240-1260.
- “The text should have used the Hardware style of literature citation, which is normally used in research papers.”
The citation style is compliant with the journal’s “Instructions to authors”:
https://www.mdpi.com/journal/sustainability/instructions#preparation
- “Finally, when citing the literature (References), the number should have been greater than the 14 mentioned.”
More references have been added throughout the text.
Thank you again for your useful comments.
Reviewer 2 Report
The article has scientific value, theoretical significance, but there are several questions and comments:
1) It should be explained what the data source is "...? vary from 0.01 to 2, and we fix all other parameter values as: a = 0.3, A= 223 0.45, ? = 0.12, ? = 2, τ=0.35 and p = 0.5..." for numerical modeling of the influence of the main parameters on the economic growth and debt. What is the origin of the accepted values of indicators?
2) It is necessary to substantiate the practical significance of the study: what is the further mechanism for using the presented theoretical developments for the real management of public debt and the development of a "green" economy? Who is the potential consumer of this study?
3) The authors note that "... n particular, as the percentage of interest paid to international creditors that is returned in the form of foreign aid to support public investment in infrastructure (p) increases, the long run steady state growth rate (?) increases, and the debt to output ratio (In/?) decreases...". Is there always a stable logical chain of "debt repayment – assistance (investment) – sustainable growth – debt-to-output ratio"? can the established relationship change trends under various external and internal conditions of state development or disturbing influences?
4) Over 50% of the literature sources used are older than 2008. Have the authors studied enough modern scientific and practical publications on the research topic?
Author Response
Dear reviewer,
First of all we would like to thank you for your time and efforts.
Your comments have indeed led to an improved paper.
On the particular comments:
- “The article has scientific value, theoretical significance, but there are several questions and comments:
1) It should be explained what the data source is "...? vary from 0.01 to 2, and we fix all other parameter values as: a = 0.3, A= 223 0.45, ? = 0.12, ? = 2, τ=0.35 and p = 0.5..." for numerical modeling of the influence of the main parameters on the economic growth and debt. What is the origin of the accepted values of indicators?”
On page 6 we have added the following references:
- Chen, Shu-Hua, and Jang-Ting Guo. "Progressive taxation and macroeconomic (In) stability with productive government spending." Journal of Economic Dynamics and Control 37.5 (2013): 951-963.
- Vella, Eugenia, Evangelos V. Dioikitopoulos, and Sarantis Kalyvitis. "Green spending reforms, growth, and welfare with endogenous subjective discounting." Macroeconomic Dynamics 19.6 (2015): 1240-1260.
While we consider the above as adequate, there are many more examples in the literature such as:
- Angyridis, C. (2015). Endogenous growth with public capital and progressive taxation. Macroeconomic Dynamics, 19(6), 1220-1239.
- Atolia, M., Awad B. and Marquis, M. (2011). ‘Linearization and higher-order
approximations: How good are they?’, Computational Economics, Vol. 38, pp. 1-31. - Barro, R., and Xavier Sala-i-Martin. "Economic growth second edition." (2004).
- Palivos, T. and Yip, C. K. and Zhang, J. (2003). ‘Transitional Dynamics and Indeterminacy of equilibria in an Endogenous Growth Model with a Public Input’, Review of Development Economics, Vol. 7, pp. 86-98.
In general, the numerical examples provided in the text are illustrations of the analytical findings of the paper that we provide in the mathematical appendices, and allow us to show how the steady state values react to changes of the policy parameters (φ and p), going from the generality of the analytical results to case specific results.
For the policy parameter φ (see equation 9) that captures the internal to eternal debt ratio, one can reasonably assume that is varies from 0.01 (e.g., a poor country with limited opportunities for domestic funding), to 2 which would be a highly developed country that has limited dependence on international funding.
There is no existing data for the policy parameter p (1>p>0), as we are adding it to the literature through this paper, but it is realistic to assume that it can vary from 0.01 (almost no foreign aid) to 1 (all interest payments are returned as foreign aid).
The examples for φ and p have been added to the text (page 6).
- “2) It is necessary to substantiate the practical significance of the study: what is the further mechanism for using the presented theoretical developments for the real management of public debt and the development of a "green" economy? Who is the potential consumer of this study?”
This is mostly for those policy makers, or those who directly support them. To clarify this, we have added a couple of lines in the introduction and a paragraph in the last section (5 Discussion), in addition to the already existing text on pages 4 and 5.
Additionally, for the mechanism of the real management, we have made further clarification sin the introduction and inserted the following in the last section: “Another consideration regards the design of the particularly financial mechanism, and the relevant institutional set-up. For a similar case one can refer to [3] which described the basic mechanics for a temporary debt standstill due to COVID-19. Although their description is relatively high level, the text demonstrates how such mechanisms can become quite complicated requiring specialised experts. For a step-by-step guide for a debt restructuring, see ….”
- “3) The authors note that "... n particular, as the percentage of interest paid to international creditors that is returned in the form of foreign aid to support public investment in infrastructure (p) increases, the long run steady state growth rate (?) increases, and the debt to output ratio (In/?) decreases...". Is there always a stable logical chain of "debt repayment – assistance (investment) – sustainable growth – debt-to-output ratio"? can the established relationship change trends under various external and internal conditions of state development or disturbing influences?”
We fully agree there are many factors that can affect this relationship and the actual impact. For example, the credibility of the mechanism, internal stability of the country, large external shocks, etc. However, these would be case specific while our proposal is more high level. We have also added in the introduction: “As external debt is in effect still serviced, and investments continuously realised, there are no significant lock-ins, especially for future political leaders. In other words, one could envision an annual or biennial evaluation of the scheme while allowing a termination by either party without loss of the benefits already realised.”
Also, for our specific model, we recall that 1>p>0 is a parameter that represents the percentage of interest payments that is returned from creditors in order to support public investment in green infrastructure. The setup of the model, the household maximization problem and the firm maximization problem along with the fiscal rules used in the model exhibit a three dimensional set of differential equations (17, 18 and 19) that fully can describe the economy along with the long run growth rate obtained in eq. 16. We can observe in the steady state analysis that the main endogenous variable that determines all other endogenous variables and economic growth is ε(t). At the steady state the variable ε(t) can be obtained by the law of motion of ε(t) (see equation A23), by setting its time derivative equal to 0 (the steady state that in Appendix B. Existence and uniqueness has been proved to exist). At this step we can observe that the value of ε(t) is determined by the parameters of the model, and also for p. In Appendix C, we show the impact of a change of p on economic growth and on Appendix E the impact of p on the debt to GDP ratio. So for the first part of the question the answer is yes.
The second part of the question is more challenging, and it contains two parts. If we assume that “external and internal conditions of state development” do not affect the structure of the model but only some parameters, our proofs (and numerical example) suggests that we will observe a change in the steady state values of the economy and the long run growth rate. On the other hand if we assume that the aforementioned conditions do change the structure of the economy (for example equation 12 or 13) then one should remodel the changes into the appropriate equations, resolve the model and investigate these relationships. In a nutshell, we cannot predict what will happen following dramatic changes, only in parameter variations.
The second part of the second part is referring to “disturbing influences”. These type of shocks refer to jumps in one or more endogenous variables. This means that the steady state of the economy is unchanged, but the new values of the parameters act as initial values of the dynamic system of differential equations (17, 18 and 19), and calls for dynamic and stability analysis, not only steady state analysis, which was beyond the scope of our work. However, this class of endogenous growth models do not exhibit Lyapunov stability, and in most cases have saddle path stability. So without a full dynamic and stability analysis one cannot answer this question, since it would also require the knowledge of the exact value of the shock in order to map the new initial condition. But, for a general attempt to answer this, most commonly the economy would be placed on a position in the phase plane that would normally follow a diverging transitional path, and only in a few cases given the existence of a stable saddle path the economy would return in its previous state (steady state). In practice, if we would observe a divergence originated by an exogenous shock then the government with the collaboration of international creditors, or even with intervention of the IMF or the World Bank, would plan an offsetting fiscal policy (see for example the case of Greece) that would cause a reverse jump back to the initial steady state or even on to the stable saddle path.
- “4) Over 50% of the literature sources used are older than 2008. Have the authors studied enough modern scientific and practical publications on the research topic?”
More recent references have been added, in addition to the seminal papers originally included.
Thank you again for your useful comments.
Reviewer 3 Report
Major comments
The paper faces a very interesting and actual issue. It is well-written and developed in a linear and clear way. However, before publication, some adjustments are needed.
First: Overall, the Introduction section describes the objectives and makes a clear overview of the paper. I suggest including brief description of methodology, summarizing main results and contributions to literature in this section.
Second: It could be give more emphasis to the goals of debt sustainability and green growth.
Third: As for financial scheme proposed, please provide a clear description of incentive scheme of both parts (international creditors and borrowers).
Fourth: Eventually, research design, assumptions, and hypotheses might be clearly state. Conclusions and results in terms of economic intuitions are not clearly presented.
Minor comments
Pag 2 line 80: please, correct typo “rear”
Pag 3 line 94-95 : please, delete the space between “the” and “change”
Pag 3 eq. (6): please, check that all terms of equations are written in a correct way as for the notation
Pag 3 line 106: check the notation
Pag 3 lines 117-120: in general, the paper is clear, but some sentences could be simplified. I suggest reviewing the English and simplifying some sentences
Pag 4 lines 151-152: please, correct errors related to references
Pag 5 line 197: please, close parenthesis
Pag 6 line 223 – 224 : please, motivate the choice of these values “we fix all other values…” “we fix all other values…”
Pag 8 line 333: please, correct the error
Pag 8 line 334: please, check and correct the notation (?)(t) and be(t)
Pag 10 line 345: please, centralize and/or split the equation
Pag 11: equations A25- A30 appear smaller than others. Please check the dimension of characters is the same for all equations of the model.
Pag 11 line 381: correct the error
Pag 12 line 410: please, correct typo in “public debt ration ratio”
Pag 13: the references are not in alphabetical order. Please check the standard and fix it.

Author Response
Dear reviewer,
First of all we would like to thank you for your time and efforts.
Your comments have indeed led to an improved paper.
On the comments:
- “The paper faces a very interesting and actual issue. It is well-written and developed in a linear and clear way. However, before publication, some adjustments are needed.
First: Overall, the Introduction section describes the objectives and makes a clear overview of the paper. I suggest including brief description of methodology, summarizing main results and contributions to literature in this section.
Fourth: Eventually, research design, assumptions, and hypotheses might be clearly state.”
We have revised the introduction and discussion to further emphasize these points.
- “Second: It could be give more emphasis to the goals of debt sustainability and green growth.”
We have provided some additional information and references throughout the introduction.
- “Third: As for financial scheme proposed, please provide a clear description of incentive scheme of both parts (international creditors and borrowers). … Conclusions and results in terms of economic intuitions are not clearly presented.”
We added some more information on the incentives in both the introduction and the discussion, complementary to the already text on pages 4 and 5.
Thank you again for your comments.
Round 2
Reviewer 1 Report
I don't have any more suggestions for authors.
Reviewer 3 Report
I don't have any more comments for authors.